# Risk of Developing Pancreatic Cancer in Patients with Chronic Pancreatitis

**DOI:** 10.3390/jcm9113720

**Published:** 2020-11-19

**Authors:** Miroslav Vujasinovic, Ana Dugic, Patrick Maisonneuve, Amer Aljic, Robin Berggren, Nikola Panic, Roberto Valente, Raffaella Pozzi Mucelli, Alexander Waldthaler, Poya Ghorbani, Maximilian Kordes, Hannes Hagström, Johannes-Matthias Löhr

**Affiliations:** 1Department of Upper Abdominal Diseases, Karolinska University Hospital, 14186 Stockholm, Sweden; nikola.panicmail@gmail.com (N.P.); alexander.waldthaler@ki.se (A.W.); poya.ghorbani@sll.se (P.G.); maximilian.kordes@sll.se (M.K.); hannes.hagstrom@ki.se (H.H.); matthias.lohr@ki.se (J.-M.L.); 2Department of Medicine, Huddinge, Karolinska Institutet, 14186 Stockholm, Sweden; ana.dugic@ki.se (A.D.); amer.aljic@stud.ki.se (A.A.); robin.berggren@stud.ki.se (R.B.); roberto.valente@ki.se (R.V.); 3Division of Epidemiology and Biostatistics, IEO, European Institute of Oncology IRCCS, 20141 Milan, Italy; patrick.maisonneuve@ieo.it; 4Department of Clinical Science, Intervention, and Technology (CLINTEC), Karolinska Institutet, 14186 Stockholm, Sweden; raffaella.pozzi-mucelli@sll.se; 5Department of Abdominal Radiology, Karolinska University Hospital, 14186 Stockholm, Sweden; 6Clinical Epidemiology Unit, Department of Medicine, Solna, Karolinska Institutet, 17177 Stockholm, Sweden

**Keywords:** pancreas, cancer, chronic pancreatitis, risk factors

## Abstract

**Background**: Patients with chronic pancreatitis (CP) have an increased risk of developing pancreatic ductal adenocarcinoma (PDAC). We present data on PDAC in one of the most extensive European single-centre cohort studies of patients with CP. **Methods**: Retrospective analysis of prospectively collected data of patients with CP was performed. Aetiology of CP was determined according to the M-ANNHEIM classification system and only patients with definite CP > 18 years at data analysis were included. The final dataset included 581 patients with definite CP diagnosed between 2003 and 2018. **Results:** At CP diagnosis, there were 371 (63.9%) males and 210 (36.1%) females (median age 57 years, range 2–86). During 3423 person-years of observation, six pancreatic cancers were diagnosed (0.2% year). The mean time between diagnosis of CP and the occurrence of PDAC was 5.0 years (range 2.7–8.6). None of the cancer patients had a family history of PDAC. Diabetes mellitus (DM) was present in five of six (83.3%) patients with PDAC: in three patients before and in two after CP diagnosis. Clinical/laboratory signs of pancreatic exocrine insufficiency (PEI) were present in five of six (83.3%) patients with PDAC: in two at diagnosis of CP and in three after diagnosis. The mean survival time was 4 months after the diagnosis of PDAC (range 0.5–13). PDAC occurred significantly more often (*p* < 0.001) in two groups of patients without previous acute pancreatitis (AP): 2 of 20 patients (10%) with low body mass index (BMI) and PEI and in 3 of 10 (30%) patients with high BMI and DM at diagnosis of CP. **Conclusions:** Patients with CP have a high risk of developing PDAC, although risk is low in absolute terms. Our data suggest the possibility of defining subgroups of patients with a particularly elevated risk of PDAC. Such a possibility would open a path to personalised decision making on initiation of PDAC surveillance of patients with no previous episode of AP, (i) with low BMI and PEI, or (ii) elevated BMI and DM.

## 1. Introduction

Chronic pancreatitis (CP) is a pancreatic disease in which recurrent inflammatory episodes replace pancreatic parenchyma with fibrous connective tissue [1]. According to a systematic review, a meta-analysis, and a meta-regression of population-based cohort studies, the global pooled incidence of CP is 10 cases per 100,000 person-years, with a significantly higher incidence in the European region than in the American region and is twice as high in men as in women [2,3]. Pancreatic ductal adenocarcinoma (PDAC) is ranked the 11th most common cancer in the world, representing 4.5% of all deaths caused by cancer in 2018 [4]. Despite some progress over the past decade, the prognosis of PDAC remains poor, with a 5-year overall survival across the whole patient population of only 9% [5]. CP’s relationship with PDAC has been the subject of intense research since the 1990s, but most studies on the risk of developing PDAC in patients with CP were biased. Major limitations are potential misclassification; possible confounding variables, including smoking and alcohol consumption; different lengths of the follow-up between CP diagnosis and the occurrence of PDAC; demographic heterogeneity of patients; and a lack of data on differences in patients with various aetiologies of CP [6]. Current guidelines on the management of patients with CP do not include recommendations on an optimal time interval and modality of PDAC surveillance, which poses a huge problem for practitioners in daily clinical practice. Lack of epidemiological data on pancreatic diseases in the general population of most parts of the world constitutes a knowledge gap and a barrier in developing an adequate strategy for prevention and effective surveillance of patients with CP [3,7]. Here, we present data on PDAC in one of the largest European single-centre cohorts of patients with CP and identify specific subgroups of patients with CP at significantly increased risk of developing PDAC.

## 2. Patients and Methods

### 2.1. Study Population

We retrospectively analysed prospectively collected patients with a diagnosis of CP. Patients were identified using codes K86.0 and K86.1 according to the 10th revision of the International Classification of Diseases (ICD-10). From 2003–2018, available electronic medical charts of all patients (historical cohort) with CP in the electronic database at the Outpatient Pancreatic Clinic at the Department of Upper Abdominal Diseases, Karolinska University Hospital, Stockholm, Sweden were analysed. We excluded patients if they were <18 years of age at the time of data analysis, without a Swedish personal identification number, with missing data in medical charts, with probable CP, and in whom PDAC occurred <2 years after diagnosis of CP. Patients diagnosed in 2019 and 2020 were excluded as well because of the short follow-up. Aetiology of CP was determined according to the M-ANNHEIM classification system [8] and only patients with definite CP were included in one or more of the following groups: alcohol, nicotine, nutritional factors, hereditary factors, efferent duct factors, immunological, and miscellaneous/other. Definite CP, according to M-ANNHEIM criteria, was diagnosed by imaging (computed tomography, magnetic resonance imaging, or both) with one or more of the following criteria: (a) pancreatic calcifications, (b) moderate or marked ductal lesions, (c) marked and persistent exocrine insufficiency defined as pancreatic steatorrhea markedly reduced by enzyme supplementation, or (d) typical histology of an adequate histological specimen [8]. Probable CP, according to the M-ANNHEIM criteria, was diagnosed with one or more of the following criteria: (a) mild ductal alterations, (b) recurrent or persistent pseudocysts, (c) pathological test of pancreatic exocrine function, or (d) endocrine insufficiency (abnormal glycated haemoglobin—HbA1c) [8]. Diagnosis of autoimmune pancreatitis (AIP) was established in accordance with the currently valid international consensus diagnostic criteria for AIP [9]. CP was considered “unexplained” when no cause could be identified.

### 2.2. Variables

For individual patients, we recorded age; sex; family history of PDAC; pancreatic exocrine insufficiency (PEI), diagnosed clinically in patients with steatorrhea or laboratory using faecal elastase with a cut-off of 200 µg/g of stool; body mass index (BMI); alcohol consumption (overconsumption was defined as an intake of ≥5 standard drinks per day, with standard drinks defined as 0.25 L of beer, 0.1 L of wine, or 4 cL of hard liqueur); smoking habits (never smoked/former smoker/active smoker); number of smoking pack-years, with one pack-year defined as 20 cigarettes smoked every day for 1 year; presence of diabetes mellitus (DM) at the time of CP diagnosis; and data on the occurrence of acute pancreatitis (AP) before the diagnosis of CP. To avoid the impact of PDAC being initially misdiagnosed as CP and possible resultant selection bias, we excluded cases of PDAC in the first 2 years of follow-up. Person-years were calculated from the date of CP diagnosis to the date of last personal contact with the patient or death, whichever occurred first.

### 2.3. Follow-Up

Follow-up was defined as the time between the date of CP diagnosis and the time of death or the time of the last contact with the patient or the time of the PDAC diagnosis. The PDAC diagnosis was confirmed on weekly multidisciplinary (surgeons, gastroenterologists, radiologists, oncologists) pancreatic conference meetings based on clinical and imaging data.

### 2.4. Statistical Analysis

Using the Kaplan Meier method (95% confidence intervals), we estimated the cumulative incidence of pancreatic cancer with the number of patients at risk at several time points. The average annual rate of pancreatic cancer—overall and according to selected patient characteristics—was calculated by dividing the number of pancreatic cancers diagnosed by the total number of person-years accumulated in the specific subgroup. The log-rank test was used to compare the risk of pancreatic cancer between groups. The analyses were performed using the SAS software version (version, 9.4 SAS Institute, Cary, NC, USA). *p*-values < 0.05 (two-sided) were considered statistically significant.

### 2.5. Ethical Considerations

The study was approved by the Regional Ethics Committee (Swedish: Regional Etikprövningsnämden) in Stockholm, Dnr: 2020-02209. The requirement for individual informed patient consent was waived by the committee owing to the nature of the study and that patients were not directly involved.

## 3. Results

From the database, the total number of patients with ICD codes for CP (K86.0 and K86.1) was 954 (Figure 1). Medical data were missing in 150 patients; thus, these patients were excluded from further analysis. Based on M-ANNHEIM criteria, there were 804 patients with confirmed CP; of these, 209 had probable CP and therefore were excluded from the final analysis. We also ruled out 14 patients for whom PDAC occurred <2 years after the diagnosis of CP.

The final dataset included 581 patients (371 [63.9%] males and 210 [36.1%] females) with definite CP diagnosed between 2003 and 2018 (Figure 1). The median age at diagnosis of CP was 57 years (range 2–86) (Table 1). During 3423 person-years of observation, six pancreatic cancers were diagnosed (0.2% year). The mean number of years between the diagnosis of CP and the occurrence of PDAC was 5.0 (range 2.7–8.6). None of the cancer patients had a family history of PDAC. DM was present in five (83.3%) patients with PDAC: in three patients before the CP and in two after the CP diagnosis. Clinical/laboratory signs of PEI were present in five (83.3%) of the patients with PDAC: in two patients at the time of CP diagnosis and in three after the CP diagnosis. Most of the patients in this subgroup (83.3%) were smokers (Table 2). Mean survival time after the PDAC diagnosis was 4 months (range 0.5–13).

Post hoc analysis revealed that PDAC occurred significantly more often (*p* < 0.0001) in two small subgroups of patients without previous AP. Two of 20 patients (10%) with low BMI and PEI at CP diagnosis and 3 of 10 (30%) with high BMI and DM at CP diagnosis developed PDAC compared to one patient in the remaining subgroup (Table 3). When we performed a sensitivity analysis and included cases of PDAC that occurred within 2 years of the CP diagnosis and when a-priori had been excluded from the analysis, the proportion of patients with PDAC in the low BMI and PEI subgroup was 9.1% and in the high BMI and DM was 16.7%, which can be compared to 1.8% in the remaining subgroup (Table 3).

## 4. Discussion

Patients with CP have an excess risk of developing PDAC but the exact association between CP and PDAC remains poorly understood [6]. In this large single-institution cohort, the incidence rate of PDAC was 0.2% annually, which is comparable to elevated rates previously reported, but is still low in absolute terms [10,11]. Overall, the cumulative incidence of pancreatic cancer is similar to that previously reported [12] (Figure 2). We found, however, that the risk of developing PDAC is unevenly distributed across the two subgroups of patients at increased risk: (i) patients with DM and a high BMI and (ii) patients with PEI and a low BMI. We hypothesise that these two subgroups represent two distinct pathomechanisms that promote the development of PDAC in patients with CP.

CP is generally associated with type 3c diabetes (T3cDM) through inflammation and fibrotic transformation of the pancreas, which impairs β-cell activity and eventually causes islet cell loss [13]. T3cDM is commonly linked to malnutrition, but a large proportion of CP patients are overweight and not deficient in micronutrients [14]. DM in these patients might resemble more closely type 2 diabetes mellitus (T2DM) with excess insulin and peripheral insulin resistance [15]. In the absence of CP, T2DM is also a risk factor for the development of PDAC and is mediated by tumour-associated factors [16]. DM regularly precedes the diagnosis of PDAC by months to years, but unlike T2DM in patients without cancer, it is associated with poor glucose homeostasis despite continued weight loss [16]. Detrimental reciprocity of concurrent DM and CP with a more than 30-fold increased risk of PDAC has previously been recognised in a population-based study [17]. However, our data suggest that a narrow subset of patients with maintained BMI within the context of DM and CP develop PDAC. A possible explanation for this observation could be several signals triggered by high insulin levels and obesity that promote proliferation, especially through the *KRAS* pathway. These signals would cause malignant transformation within the context of chronic inflammation in the pancreas [18,19,20]. 

PEI is a hallmark of CP, which affected nearly half of our cohort patients. It is caused by alterations of the pancreatic anatomy and impaired regulation of the pancreatic secretion, which are also typical of pancreatic cancer [21]. A low BMI at diagnosis, in contrast, occurred only in 16.2% of the CP patients in this study, although weight loss is the natural consequence of PEI associated with PDAC [22,23]. Molecular changes in the pancreas occur years before malignant tumour occurrence [10] and metabolic changes begin to occur as early as 30 months before a PDAC diagnosis [24]. Tissue breakdown with sarcopenia leading to cachexia (low BMI) can occur long before the diagnosis of clinically overt pancreatic cancer, as based on intrinsic tumoral factors [22]. Because the development of clinically overt PDAC was calculated to take up to 10 years, [10] incipient pancreatic cancer may contribute to this wasting in the years preceding the diagnosis [23]. Weight loss manifests typically 6 to 18 months before clinical PDAC [24]. We therefore conclude that low BMI and PEI in patients with CP define a high-risk population with latent PDAC. This interpretation could also explain the sharp decrease in PDAC risk between 2 and 5 years in a meta-analysis of cohort and case-control studies of CP and PDAC as clinical symptoms in patients with latent disease debut during this period [6]. 

The retrospective design of the study and a low absolute number of patients with PDAC and the fact that we observed clustering of these cases in post hoc analysis are the major limitations of this study and warrant confirmation in additional studies. To reduce bias from inclusion of patients without CP, we limited this study to definite CP based on the M-ANNHEIM criteria. Following the common practice in other epidemiological studies on the association of CP and PDAC [6], we excluded all patients for whom PDAC occurred in the first 2 years after the diagnosis of CP to limit detection bias. However, we conducted a sensitivity analysis that included these cases after PDAC was observed in specific subgroups. Screening for PDAC in all patients with CP for a more extended period is not feasible because of the large number of patients with CP, the low incidence of PDAC, and the lack of non-invasive diagnostic procedures. However, given the number of PDAC in the first 2 years after the CP diagnosis, it is an open question whether it is reasonable to follow all CP patients carefully in this short period. Patients with hereditary pancreatitis and mutations in the cationic trypsinogen gene are the exception because they have a remarkably increased risk of developing pancreatic cancer [11] and should be followed on a yearly basis [1]. The lifetime risk for developing PDAC in patients with hereditary pancreatitis is approximately 30% at 70 years of age [25], which is considerably higher compared to other CP aetiologies. Another group that may pose a higher risk for developing PDAC is patients with AIP, especially AIP type 1 as a part of immunoglobulin G4-related disease (IgG4-RD) [26,27]. IgG4-RD, and particularly AIP, could be linked to an increased risk of developing malignant disease relative to the general population. Consequently, current European guidelines recommend lifelong surveillance in patients with IgG4-RD [28]. The overall demographic characteristics, exposures, and outcomes of our cohort are comparable to other studies in the field (Table 4). Note, however, that we found a higher proportion of female patients (36.1%) than in similar studies [12,29,30,31,32,33,34,35,36,37,38,39,40,41,42], probably because of the higher number of patients with CP of non-alcoholic aetiology. Still, confounding variables, such as smoking and alcohol consumption, and two independent risk factors of both CP and PDAC remain general problems of association studies of CP and PDAC [12,43,44]. Accordingly, five of six (83.3%) patients with PDAC in our group were either active or former smokers.

## 5. Conclusions

Patients with well-defined CP have an excess risk of PDAC. Patients with DM and a high BMI or PEI and a low BMI at diagnosis of CP may constitute specific risk groups among all patients with CP and should therefore be studied in independent populations. Identifying such risk groups may overcome practical limitations of screening for PDAC in all patients with CP given the large number of patients with CP, the low incidence of PDAC, and the lack of non-invasive diagnostic procedures.

## Figures and Tables

**Figure 1 jcm-09-03720-f001:**
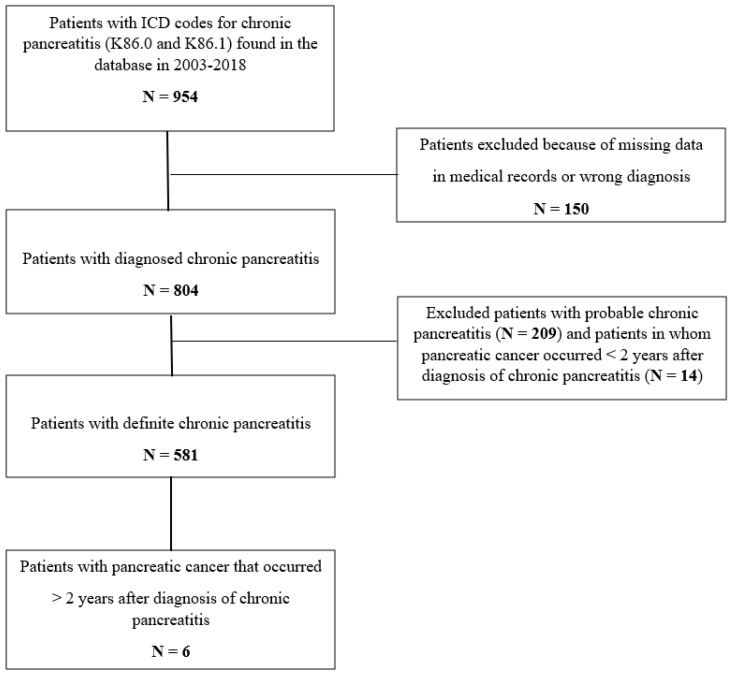
Flow chart of patients.

**Figure 2 jcm-09-03720-f002:**
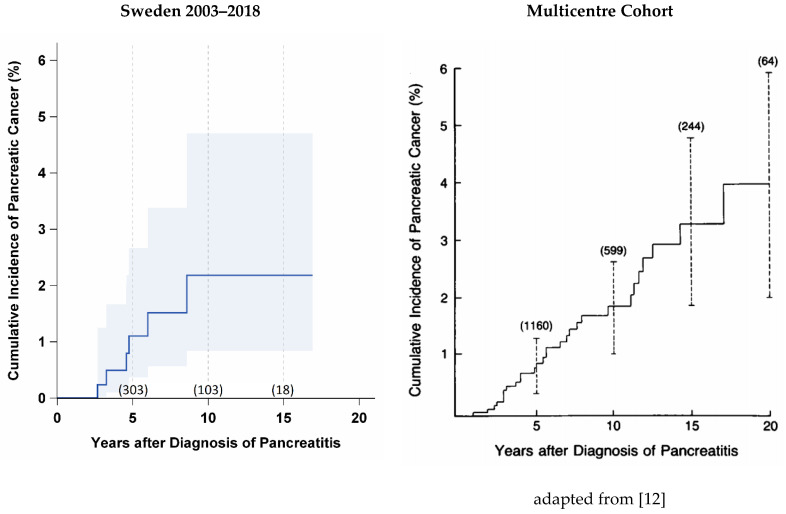
Cumulative incidence of pancreatic cancer in Sweden compared to the multicentre cohort study. The number in parenthesis indicates the number of patients at risk at different time points.

**Table 1 jcm-09-03720-t001:** Patient characteristics.

Characteristics	All CP Patients *N* (%) ^†^	Pancreas Cancer *N* (%)	Person-Year	Rate Per 100-Year	Log-Rank *p*
All	581 (100.0)	6 (1.03)	3423	0.18	
Sex					
Female	210 (36.1)	3 (1.43)	1232	0.24	
Male	371 (63.9)	3 (0.81)	2191	0.14	0.48
Age at diagnosis					
<50	183 (31.5)	-	1223	0.00	
50–59	134 (23.1)	-	842	0.00	
60–69	133 (22.9)	4 (3.01)	710	0.56	
70–79	105 (18.1)	2 (1.90)	532	0.38	
≥80	26 (4.5)	-	116	0.00	0.020
BMI at diagnosis					
<20	94 (16.2)	2 (2.13)	554	0.36	
21–25	165 (28.4)	-	941	0.00	
26–31	98 (16.9)	3 (3.06)	492	0.61	
31+	18 (3.1)	1 (5.56)	93	1.08	0.065
Family history of pancreas diseases					
No	404 (69.5)	4 (0.99)	2491	0.16	
Yes	41 (7.1)	-	258	0.00	0.51
Aetiology					
Idiopathic	47 (8.1)	-	229	0.00	
Alcohol and nicotine	220 (37.9)	1 (0.45)	1343	0.07	
Nicotine	81 (13.9)	2 (2.47)	458	0.44	
Alcohol	49 (8.4)	1 (2.04)	312	0.32	
Hereditary	38 (6.5)	-	281	0.00	
Immunological	59 (10.2)	-	318	0.00	
Efferent duct	62 (10.7)	1 (1.61)	359	0.28	
Miscellaneous/other	25 (4.3)	1 (4.00)	123	0.81	0.40
Diabetes mellitus at diagnosis					
No	419 (72.1)	3 (0.72)	2570	0.12	
Yes	147 (25.3)	3 (2.04)	799	0.38	0.12
PEI at diagnosis					
No	273 (47.0)	3 (1.10)	1749	0.17	
Yes	226 (38.9)	3 (1.33)	1082	0.28	0.52
History of acute pancreatitis					
No	195 (33.6)	5 (2.56)	986	0.51	
Yes	379 (65.2)	1 (0.26)	2376	0.04	0.003
Recurrent acute pancreatitis					
No	303 (52.2)	6 (1.98)	1705	0.35	
Yes	267 (46.0)	-	1646	0.00	0.014

^†^ Percentages do not add-up to 100 because information is missing for some patients: body mass index = BMI (*n* = 206), family history (*n* = 136), symptoms (*n* = 30), calcifications (*n* = 15), diabetes (*n* = 15), pancreatic exocrine insufficiency = PEI (*n* = 82), acute pancreatitis (*n* = 7), recurrent acute pancreatitis (*n* = 11). CP = chronic pancreatitis.

**Table 2 jcm-09-03720-t002:** Demographic and clinical characteristics of patients in whom pancreatic ductal adenocarcinoma occurred more than 2 years after the diagnosis of chronic pancreatitis.

Patient	Age *	Sex	Aetiology CP	Previous AP	Smoking	Alcohol	BMI	DM	PEI	Time to PDAC **	Survival after PDAC
1	63	female	nicotine	no	former	never	18.9	developed after CP diagnosis; (1 year before PDAC diagnosis)	present at CP diagnosis	2.7 years	1.5 months
2	65	male	alcohol and nicotine	no	active	30 years	18.0	developed after CP diagnosis; (at the time of PDAC diagnosis)	presentat CP diagnosis	6.6 years	6 months
3	64	male	unexplained	no	former	never	26.3	present at CP diagnosis; (52 years before PDAC diagnosis)	developed after CP diagnosis	8.6 years	13 months
4	68	female	alcohol and nicotine	no	former	40 years	32.4	present at CP diagnosis; (33 years before PDAC diagnosis)	developed after CP diagnosis	7.0 years	1 month
5	74	female	nicotine	no	active	never	26.0	present at CP diagnosis; (11 years before PDAC diagnosis)	developed after CP diagnosis	6.0 years	2 months
6	79	male	efferent duct factors	yes	never	never	29.1	no	no	4.8 years	0.5 months

CP= chronic pancreatitis; AP = acute pancreatitis; BMI = body mass index at the time of CP diagnosis; DM = diabetes mellitus; PEI = Pancreatic exocrine insufficiency; PDAC = pancreatic ductal adenocarcinoma; *= age at the time of chronic pancreatitis diagnosis; **= period between the diagnosis of chronic pancreatitis and diagnosis of pancreatic cancer.

**Table 3 jcm-09-03720-t003:** Pancreas cancer in patients with specific characteristics at diagnosis of CP.

	Patients *N*	Pancreatic Cancers Diagnosed in the First 2 Years after CP * *N* (%)	Pancreas Cancers Diagnosed More than 2 Years after CP *N* (%)	
All	595	14 (2.4)	6 (1.01)	
**Risk group**				
No previous AP, low BMI, and PEI	22	2 (9.1)	2 (9.1)	
No previous AP, high BMI, and DM	12	2 (16.7)	3 (25.0)	
Other CP patients	561	10 (1.8)	1 (0.18)	*p* < 0.001

* These patients were excluded from the main analysis. AP = acute pancreatitis; CP = chronic pancreatitis, BMI = body mass index; PEI = pancreatic exocrine insufficiency; DM = diabetes mellitus.

**Table 4 jcm-09-03720-t004:** Cumulative incidence of pancreatic cancer in cohorts of patients with chronic pancreatitis.

First Author	Year	Study Type	Period	Follow-Up	*N*	PDAC	SEX	AGE (Years)
Rocca [29]	1987	Single centre study	Italy 1970–1984	14 years	172	2 cases (1.1%)	males 85.7%	50
Lowenfels [12]	1993	Multicentric multinational	Multinational 1946–1989	7.4 years (11,438-person years)	1552	29 cases * (2.5%)	males 81%	44.6
Karlson [30]	1997	Swedish Inpatient Registry	Sweden 1965–1983	NA	4546	189 cases (4.2%)	males 72%	53.2
Talamini [31]	1999	Single centre study	Italy 1971–1995	10 years (**7287**-person years)	715	14 cases (2.0%)	males 88%	40.8
Malka [32]	2002	Single centre study	France 1973–1997	9.2 years	373	4 cases (1.1%)	males 86%	42
Seicean [33]	2006	Single centre study	Romania 1999–2005	2 years	82	3 cases (3.6%)	Male:female Ratio = 6.5:1	48.7
Goldacre [34]	2008	Oxford Record Linkage Study	United Kingdom 1963–1999	NA	1496	86 cases (5.7%)	males 55%	NA
Pedrazzoli [35]	2008	Single centre study	Italy 1970–1997	NA	170	2 cases (1.2%)	males 87%	45
Wang [36]	2011	Single centre study	China 1997–2007	8.5 years	420	4 cases (0.9%)	males 68%	43.4
Kudo [37]	2011	Single centre study	Japan 1970–2008	NA	218	9 cases (4.1%)	males 91.3%	56.8
Dite [38]	2012	Single centre study	Czech Republic 1992–2005	6.7 years	223	13 cases (5.8%)	males 66.4%	56.1
Ueda [45]	2013	Nationwide survey	Japan 2009–2010	8 years	506	19 cases (3.7%)	males 83.6%	52.5
Bang [39]	2014	Nationwide register retrospective cohort	Denmark 1997–2010	71,814-person years	11,972	510 cases (4.25%) 172 cases * (1.4%)	males 66.5%	54.5
Hao [40]	2017	Single centre study	China 2000–2013	8.0 years	1656	21 cases (1.3%)	males 69.6%	43
Zheng [41]	2019	Single centre study	China 2009–2017	4.4 years	650	12 cases (1.8%)	males 78.5%	45
Agarwal [42]	2020	Single centre study	India 1998–2019	3.6 years	1415	29 cases (2.0%)	males 78.3%	34
Present Study	2020	Single centre study	Sweden 2003–2018	5.9 years (3423-person years)	581	6 cases * (1.0%) 0.2% per year	males 63.9%	55.6

*N* = number of patients included in the study; CP = chronic pancreatitis; PDAC = pancreatic ductal adenocarcinoma; NA = not available. * 2 years after diagnosis of CP.

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
