# Peer review of "Risk of Developing Pancreatic Cancer in Patients with Chronic Pancreatitis"

_jcm, 2020, doi:10.3390/jcm9113720_

Round 1

Reviewer 1 Report

Retrosepctive study regaring risk on PDAC in patients with CP

Introduction

Clearly describes aim

Methods/Results

- 'The PDAC diagnosis was confirmed on weekly multidisciplinary (surgeons, gastroenterologists, radiologists, oncologists) pancreatic conference meetings based on clinical and imaging data.
PDAC not histology confirmed? please comment

- PEI and DM were not present in 38.9 and 25%, repsectively. Was faecal elastase and glc measured in the remaining cases of was it missing in majority of patients?

Discussion

Clearly written

Author Response

Comment: - 'The PDAC diagnosis was confirmed on weekly multidisciplinary (surgeons, gastroenterologists, radiologists, oncologists) pancreatic conference meetings based on clinical and imaging data.'

PDAC not histology confirmed? please comment

Authors’ answer: Thank you for the comment! Two patients undergone surgery and histopathology confirmed diagnosis of pancreatic cancer. In four patients, diagnosis was confirmed on typical imaging and surgery was not performed due to advanced pancreatic cancer involving large vessels and liver metastasis (in these patients only best supportive care was indicated, and the biopsy was not performed). In table 2 we mentioned survival in patients, but we did not go deeply in their diagnosing process and treatment because it was out of scope of this manuscript.

Comment: - PEI and DM were not present in 38.9 and 25%, respectively. Was faecal elastase and glc measured in the remaining cases or was it missing in majority of patients?

Authors’ answer:

Thank you for the comment! In legend of table 1 we explained that percentages do not add-up to 100 because information is missing for some patients: DM (n=15), PEI (n=82). PEI was diagnosed clinically in patients with steatorrhea or laboratory using faecal elastase with a cut-off of 200 µg/g of stool. Data were missing in 82 patients (in these patients we have neither clinical information nor elastase). The reason why we included these patients is the fact that our primary outcome in this study was to determine occurrence of pancreatic cancer and we included all patients in whom this information (cancer) was available. Secondary aim was to make association with PEI and DM and we included patients in whom these information were available. We agree with your comments and comments of reviewer number 2 who wrote that the retrospective design of the study, a low absolute number of patients with PDAC, and the observation of clustering of these cases in post hoc analysis constitute major limitations. We hope that our next study on higher number of patients from other centers will give us answer on this question.

Reviewer 2 Report

In this study, the authors evaluated the risk of pancreatic cancer in chronic pancreatitis (CP) patients.

This study includes remarkably large number of patients (as many as 581 patients) and long-term analysis (3423 person-years of observation). Although the risk of pancreatic cancer in the present study is relatively low, the authors suggested the several risk factors (no previous episode of AP, low BMI and PEI or elevated BMI and DM).

This study provides the important clue to predict the risk of pancreatic cancer. I mentioned several points.

  1. In Page 4; Mean survival time after the PDAC diagnosis was 4 months (range 0.5-13).

The prognosis of PDAC was extremely poor. Were all the patients diagnosed as unresectable? Are there any patients who underwent chemotherapy, chemoradiation or surgery? Please explain the stage and the treatment for PDAC patients.

  1. The most important discussion point for PDAC is to improve the prognosis. Please discuss the strategy to improve the poor prognosis in the PDAC patients with CP because the prognosis of PDAC patients in the current study was extremely poor.

Author Response

Comment: 1. In Page 4; Mean survival time after the PDAC diagnosis was 4 months (range 0.5-13). The prognosis of PDAC was extremely poor. Were all the patients diagnosed as unresectable? Are there any patients who underwent chemotherapy, chemoradiation or surgery? Please explain the stage and the treatment for PDAC patients.

Authors’ answer: Thank you for the comment! Two patients undergone surgery and in four patients, diagnosis was confirmed on typical imaging and surgery was not performed due to advanced pancreatic cancer involving large vessels and liver metastasis (in these patients only best supportive care was indicated). In table 2 we mentioned survival in patients, but we did not go deeply in their diagnosing process and treatment because it was out of scope of this manuscript.

Comment: 2. The most important discussion point for PDAC is to improve the prognosis. Please discuss the strategy to improve the poor prognosis in the PDAC patients with CP because the prognosis of PDAC patients in the current study was extremely poor.

Authors’ answer: Thank you for the comment! We absolutely agree with you, this is crucial. Unfortunately, in presenting study, we did not find any relevant information on improving the prognosis in patients with pancreatic cancer. We only found possible risk factors that we emphasized (no previous episode of AP, low BMI and PEI or elevated BMI and DM). However, the sample is too small for strong conclusions and that was the reason why we were careful in the discussion.

Reviewer 3 Report

In this study the authors present data on pancreatic ductal adenocarcinoma (PDAC) in one of the most extensive European single-center cohort studies of patients with chronic pancreatitis (CP). Retrospective analysis of prospectively collected data of patients with CP was performed and the authors found that patients with CP have a high risk of developing PDAC, although risk is low in absolute terms. Their data suggest the possibility to define subgroups of patients with an elevated risk for PDAC, thus providing a means for personalized decision making on initiation of PDAC surveillance of patients with no previous episode of AP. These patient subgroups consist of patients with (i) low BMI and pancreatic exocrine insufficiency or (ii) elevated BMI and diabetes mellitus.

This study is well-executed and the results are clearly defined. These findings represent a good contribution to the pancreatic cancer field. However, as the authors note in the discussion, the retrospective design of the study, a low absolute number of patients with PDAC, and the observation of clustering of these cases in post hoc analysis constitute major limitations.

Author Response

Comment: This study is well-executed, and the results are clearly defined. These findings represent a good contribution to the pancreatic cancer field. However, as the authors note in the discussion, the retrospective design of the study, a low absolute number of patients with PDAC, and the observation of clustering of these cases in post hoc analysis constitute major limitations.

Authors’ answer: Thank you for the comment! We agree with your observation.